# Non-invasive specimen collections for *Mycobacterium tuberculosi*s detection in free-ranging long-tailed macaques (*Macaca fascicularis*)

Suthirote Meesawat[1,2], Nalina Aiempichitkijkarn[3], Saradee Warit[4], Mutchamon Kaewparuehaschai[5], Suchinda Malaivijitnond[2,6]*

1 Faculty of Science, Biological Sciences Program, Chulalongkorn University, Bangkok, Thailand, 2 Faculty of Science, Department of Biology, Chulalongkorn University, Bangkok, Thailand, 3 Animal Behavior Graduate Group, University of California, Davis, California, United States of America, 4 Industrial Tuberculosis Team (ITBT), IMBG, BIOTEC, NSTDA, Thailand Science Park, Pathumthani, Thailand, 5 Department of National Parks, Wildlife Rescue Center No.2 (Krabokkoo), Wildlife and Plant Conservation, Chachoengsao, Thailand, 6 National Primate Research Center of Thailand-Chulalongkorn University, Saraburi, Thailand

* Suchinda.m@chula.ac.th

**Data Availability Statement:** All relevant data are within the paper and its Supporting information files.

## Abstract

Surveillance of infectious diseases in free-ranging or wild animals has been widely conducted in many habitat-range countries after the COVID-19 episode. Thailand is located in the center of the distribution range of long-tailed macaques (*Macaca fascicularis*; *Mf*) where the animals have both frequent human contact and a high prevalence of human tuberculosis. For the large-scale detection of *Mycobacterium tuberculosis* complex (MTBC) using IS*6110*-nested PCR in free-ranging *Mf*, non-invasive sampling was developed using oral (via rope bait) and fecal (direct swabs of fresh feces) specimen collection. Firstly, the MTBC-IS*6110*-nested PCR was validated in non-invasively collected specimens, in terms of its specificity and sensitivity, and then compared with those of the invasively collected oral and rectal swabs in 24 captive MTBC-suspected *Mf*. After validation, these methods were applied to survey for the prevalence of shed MTBC (MTBCS) in four previously reported MTBC-infected populations. A total of 173 baited rope specimens and 204 freshly defecated excretions were collected. The limit of detection of the IS*6110*-nested PCR technique was 10 fg/µL and the 181-bp PCR amplicon showed 100% sequence similarity with the MTB H37Rv genome sequence. Comparing the MTBCS detection between the invasive and non-invasive collected specimens in captive suspected *Mf* revealed a significant correlation between the two types of oral specimens (oral swabs and baited ropes; $n = 24$, $r^2 = 1$, p-value < 0.001), but fresh fecal swabs showed higher MTBCS frequencies than the rectal swabs. Moreover, the proportion of MTBCS-positive free-ranging *Mf* were significantly higher in the fresh fecal swabs (8.82%; 95% CI; 4.9–12.7%) than in the baited ropes (5.20%; 95% CI; 1.9–8.5%). This result indicates that oral sampling via baited ropes and fecal sampling via defecated excretion swabs can serve as ancillary specimens for MTBCS detection in free-ranging non-human primates.

**Funding:** This research was supported by the Research Fund Senior Scholar (grant number RTA6280010 to SM), the TSRI Fund (to SM), the NSRF via the Program Management Unit for Human Resources & Institutional Development, Research and Innovation (grant number B05F640122 to SM), the 90th Anniversary of Chulalongkorn University Ratchadaphiseksomphot Endowment Fund (to SM), and Chulalongkorn University - NSTDA Doctoral Scholarship (to SM). The funders had no role in study design, data collection and analysis, decision to publish, or preparation of the manuscript.

**Competing interests:** The authors have declared that no competing interests exist.

## Introduction

During the past few years, the zoonotic transmission of emerging (i.e., SARS-CoV2) and re-emerging (i.e., monkeypox) infectious diseases has become a global focus for public health. The spread of diseases across the borders of countries, especially to developed countries, such as the United States, are mostly transmitted from imported infected wildlife or from infected travellers returning to their home countries. Although tuberculosis is a centuries-old infectious disease, it has re-emerged in many countries [1,2] and is currently the world's deadliest infectious disease, second only to COVID-19 [3].

The challenge for tuberculosis eradication is its complication of the latent or subclinical stage. Although humans are the main reservoir hosts of *Mycobacterium tuberculosis* bacterium (MTB) – a causative agent of tuberculosis, the spillback of the MTB from humans to non-human primates (NHPs) is reported in areas where free-ranging NHPs have frequent human contact and a high human tuberculosis prevalence [4,5]. The MTB complex (MTBC)-infected NHPs could form a novel animal reservoir of MTB, making the pathogens more difficult to eradicate. Thus, surveillance of MTBC infections in habitat range countries where NHPs live has been widely conducted during the past 10 years, such as in macaque monkeys in Indonesia, Thailand, Singapore, Nepal, and Gibraltar [5]; chimpanzees and baboons in Tanzania [6]; and long-tailed macaques in Thailand [7].

Specimen collection from free-ranging or wild macaques for MTB detection is a logistical challenge and needs a lot of investment (financial support, manpower, and time-consumption). Because tuberculosis is spread from one animal to another through the air and generally affects the respiratory tract, the macaques need to be anesthetized before invasive specimen collections, such as bronchoalveolar lavage or oral-pharyngeal swabs, which can be dangerous to both the macaques and human handlers. Besides, the macaques are also extremely intelligent and quickly learn how to avoid capture. In these situations, non-invasive specimen collection is a viable option [8]. Since the MTBC are shed (MTBCS) in the sputum and subsequently swallowed and excreted in feces during active infection, fecal MTBCS DNA detection was developed in free-living chimpanzees and baboons using PCR amplification of insertion sequence *6110* (IS*6110*) [6]. Although this method is practical for free-ranging NHPs, it is unrealistic if the surveillance is needed for targeted animals because the researcher needs to follow the animals until they defecate. Thus, alternative types of specimens that can be non-invasively collected and used for MTBC(S) determination in free-ranging NHPs is required.

Previously, researchers succeeded in detecting MTBCS in oral swab specimens from macaque monkeys, including long-tailed macaques (*Macaca fascicularis*), while the animals were anesthetized during invasive specimen collections [5,9]. Although various methods of non-invasive oral specimen collection in NHPs have been developed, they were mainly used for hormonal assays [10–15], genetic analysis [16,17], and more recently for surveillance of other infectious diseases, i.e., rhesus cytomegalovirus (RhCMV) and simian foamy virus (SFV), in free-ranging olive baboons (*Papio anubis*), red-tailed guenons (*Cercopithecus ascanius*), L'Hoest's monkeys (*Cercopithecus lhoesti*), and rhesus macaques (*Macaca mulatta*) [18]. Up to now, there have been no reports of non-invasive oral specimen collection for MTBC determination in free-ranging NHPs. Here, we modified the established rope bait method for oral specimen collection [17] and evaluated if the collected specimens could be used to determine tuberculosis (as MTBCS) in free-ranging long-tailed macaques living in Thailand using PCR amplification of IS*6110*. We also applied the storage condition of oral specimen preservation, which was previously used for population genetic analysis and is practical for field study, to this tuberculosis screening. The percentage of MTBCS-positive samples between non-invasively collected fecal and oral specimens was also compared.

## Materials and methods

### Animal subjects and invasive and non-invasive oral and fecal specimen collections

Twenty-four long-tailed macaques who had a history of being housed in the same area (less than 10 m apart) or in the same gang cage with MTB-infected monkeys [19] were recruited for specimen collections and validation of the MTBCS-detection using IS*6110*-nested PCR. These animals were comprised of 19 males and five females, aged over 7 years old, and of 3.2–8.0 kg body weight. They were housed in gang cages (4 x 10 x 4 m for W x L x H) exposed to natural environmental conditions at Krabok-Koo Wildlife Rescue Center (KBKWRC), Chachoengsao province, eastern Thailand and were fed with fruits and cooked rice twice a day. The monkeys were anesthetized with a mixture of Zoletil (3–5 mg per kg) and dexmedetomidine hydrochloride (0.03–0.05 mg per kg). After attaining deep anaesthesia, oral and rectal swabs were taken at the bulge of monkey's cheek pouches and anus, respectively, using a cotton swab (CITOS-WAB, China), and this day is counted as Day-1 of the study. The swabs were kept in 1.5 mL of sterile lysis buffer [0.5% (w/v) SDS, 100 mM EDTA pH 8.0, 100 mM Tris-HCl pH 8.0, and 10 mM NaCl] [20,21] at room temperature (RT) until DNA extraction. The animals were transferred to individual cages (60 x 70 x 85 cm for W x L x H) for recovery.

After animals were fully recovered, non-invasive fecal and oral specimens were collected for three consecutive days (Day-2, 3, and 4). The fecal specimen was freshly defecated excretion in the tray placed under each individual cage. A cotton swab (CITOSWAB, China) was used to swab inside the feces and this was repeated three times. The swabs were kept in 1.5 mL of sterile lysis buffer as mentioned above at RT until DNA extraction. The oral specimen was collected by the rope-bait method previously used in stump-tailed macaques (*M. arctoides*) for paternity testing [17]. Briefly, polyester ropes (6 mm in diameter; Takagi Corporation, Kagawa, Japan) were cut into 10-cm long pieces, autoclaved, dried, soaked in 20% (w/v) sucrose in 50-mL tube for a few hours, and directly provided to animals. After the animals had taken the rope, chewed it until the sweetness had gone, and discarded the rope onto the tray under the cages, the rope was then quickly collected and preserved in 3 mL of sterile lysis buffer as mentioned above. On Day-5, animals were anesthetized again, oral and rectal swabs were collected for the second round, and the animals released back to their home gang-cage. In total, there were three replications of non-invasive specimen collection and two replications of invasive specimen collection for MTBCS detection using IS*6110*-nested PCR (see below). The PCR results obtained from the invasive and non-invasive collected specimens were compared.

### DNA extraction and MTBCS detection using IS6110-nested PCR

The swab samples were incubated at 70°C for 1 h with 50 μL of 30 mg/mL lysozyme solution (Cat. No. 34388, SERVA, Germany). Genomic DNA was then extracted using a Virus/Pathogen Mini Kit (Cat. No. 937055, QIAGEN, Germany) and an automated QIAsymphony® (QIAGEN, Germany). The extracted DNA was amplified using nested PCR with IS*6110* specific primers, which was modified from that developed for human specimens [22]. In brief, the MTB-specific IS*6110* [23] was amplified for two rounds. For the first round, the Tb 294 (5'-GGACAACGCCGAATTGCGAAGGGC-3') and Tb 850 (5'-TAGGCGTCGGTGACA−AAGGCCACG-3') primers were used to yield a 580-bp amplicon. For the second round (nested) PCR, the Tb 505 (5'-ACGACCACATCAACC-3') and Tb 670 (5'-AGTTTGGTCATCAGCC-3') primers were used to yield an amplicon size of 181 bp. The 25-μL PCR reaction mixture consisted of 17.4 μL deionized distilled water, 0.5 μL of each primer (0.4 pmol/μL), 2 μL DNA template, and 4.6 μL

TaKaRa Ex Taq® Hot Start Version Kit, 2.5 μL 10X Ex Taq Buffer (Mg$^{2+}$ plus), 2.0 μL of 2.5 mM each deoxynucleoside triphosphate (dNTP), and 0.125 μL TaKaRa Ex Taq Hot (TAKARA BIO INC., Japan). The DEPC treated water was used as a negative control.

The PCR reaction was run in Verti™ 96-Well Thermal Cyclers (Applied Biosystems by Thermo Fisher Scientific, Singapore). The thermal cycling for the first round PCR was 98˚C for 1 min and then 30 cycles of 93˚C for 20 s, 65˚C for 30 s, and 72˚C for 1 min, and then followed by a final 72˚C for 10 min. That for the second round, using 1 μL of the first-round PCR product, was: 98˚C for 1 min; 30 cycles of 93˚C for 20 s, 48˚C for 30 s, and 72˚C for 30 s, and then followed by a final 72˚C for 10 min. The 217-bp ß-actin fragment of long-tailed macaques was used as a house-keeping gene in the PCR amplification and was amplified using the 5'-ATCATGTTTGAGACCTTCAACACC-3' and 5'-TGAGGATCTTCATGAGGTAGTCAG-3' primers [24]. Thermal cycling of ß-actin was similar to that in the second round of the MTBCS reaction, except that 60˚C was used as the annealing temperature.

The PCR products were visualized using 2% (w/v) agarose gel electrophoresis and stained with SYBR® Safe DNA gel stain (Lot no. 1771565, Invitrogen, USA) under UV light using a NaBI Instrument with NEOimage program (Nucleic acid Gel Imaging System, NeoScience, Korea). A 100-bp HyperLadder™ (Cat. No: BIO-33056, BIOLINE, USA) was used as a molecular standard marker.

## Specificity and sensitivity of MTBCS detection using IS6110 nested PCR

Invasively collected oral and fecal specimens that showed MTBCS-positive results were selected for validation of the specificity of the nested PCR with the IS*6110* specific primers for MTB detection. The 181-bp nested PCR products of the MTBCS-positive monkeys (four oral specimens and one fecal specimen; see Table 1) were purified using the GenUP™ Exo Sap Kit (Biotechrabbit, Germany), and submitted to the Macrogen Company (Seoul, South Korea) for DNA sequencing. These nucleotide sequences were aligned with the published MTB sequences accessed from the NCBI database using the MEGA-X Software [25]. The IS*6110*-nested PCR protocol was used in the next step of the study if the obtained amplicon sequences of our macaques showed 100% homology with that of the published MTB IS*6110* sequences.

To determine the sensitivity of MTBCS-nested PCR technique, genomic DNA of the MTB H37Rv (ATCC27294) strain was used and serially diluted (1:10) from 1 μg to 1 fg in pooled lysis buffer solution containing MTB-negative oral or fecal samples. The lowest concentration of the MTB H37Rv that could be detected by nested PCR was designated as the limit of detection (LOD), or level of sensitivity, of this technique.

## Survey of MTBCS prevalence in non-invasively collected specimens of free-ranging long-tailed macaques in Thailand

After MTBCS detection using IS*6110* nested PCR in the non-invasively collected oral and fecal specimens was validated and showed comparable results with those of the invasive collected specimens, the survey of MTBCS prevalence in free-ranging long-tailed macaques in Thailand was conducted. One common long-tailed macaque (*Macaca fascicularis fascicularis*) population at Wat Kao Thamon (WKT, GPS: 13˚02'N, 99˚57'E), western Thailand, and three Burmese long-tailed macaque (*M. fascicularis aurea*) populations at Tham Pra Khayang (TPK, GPS: 10˚19'N, 98˚45'E), World War Museum (WWM, GPS: 10˚10'N, 98˚43'E), and Mangrove Forest Research Centre (MFRC, GPS: 9˚87'N, 98˚60'E) in south-western Thailand were the subjects of this study. WKT monkeys lived in the forest patch connected to the temple, they

**Table 1. Detection of MTBCS in long-tailed macaques, by sample type collection.–and + stand for negative and positive MTBCS results, respectively.**

| No. | Monkey ID code | Invasive specimen collection | | Non-invasive specimen collection | |
|---|---|---|---|---|---|
| | | Oral swab | Rectal swab | Rope bait | Defecated excretion |
| 1 | KBK035 | - | - | - | - |
| 2 | KBK036 | - | - | - | - |
| 3 | KBK037 | - | - | - | - |
| 4 | KBK038 | - | - | - | - |
| 5 | KBK039 | - | - | - | - |
| 6 | KBK040 | - | - | - | - |
| 7 | KBK041 | - | - | - | - |
| 8 | KBK042 | - | - | - | - |
| 9 | KBK043 | - | - | - | - |
| 10 | KBK047 | - | - | - | - |
| 11 | KBK055 | - | - | - | - |
| 12 | KBK092 | - | - | - | - |
| 13 | KBK093 | - | - | - | - |
| 14 | KBK095 | - | - | - | - |
| 15 | KBK096 | - | - | - | - |
| 16 | KBK098 | - | - | - | - |
| 17 | KBK102 | + | - | + | + |
| 18 | KBK103 | + | + | + | + |
| 19 | KBK105 | - | - | - | - |
| 20 | KBK120 | - | - | - | - |
| 21 | KBK195 | + | - | + | + |
| 22 | KBK199 | + | - | + | - |
| 23 | KBK294 | - | - | - | - |
| 24 | KBK298 | - | - | - | - |
| | Total | 4/24 (16.67%) | 1/24 (4.17%) | 4/24 (16.67%) | 3/24 (12.50%) |

frequently roamed to the temple ground for human-fed-foods. All three locations of Burmese long-tailed macaques were mangrove forests, and the animals were occasionally accessed to the natural trails for human-fed-foods. Previously, these four populations were captured-and-released, anesthetized, oral and rectal swabs collected, and screened for MTBCS using IS*6110* nested PCR. This revealed MTBCS frequencies in the WKT, TPK, WWM, and MFRC populations of 9.7% (7/72), 2.5% (1/40), 1.9% (1/52), and 11.4% (4/35), respectively, for oral swab specimens, and 1.4% (1/72), 7.5% (3/40), 7.7% (4/52), and 0% (0/35), respectively, for rectal swab specimens [9].

Non-invasive specimen collections were performed when monkeys appeared on temple ground (for WKT population) or natural trails (for TPK, WWM and MFRC populations). Using the rope bait method (Fig 1), approximately 50 sucrose-soaked ropes were scattered on the ground when we saw monkeys. After the animals had taken the rope, chewed it until the sweetness had gone, and discarded the rope onto the ground, the rope was then quickly collected. After all the baited-rope specimens were collected, animals were followed. Once they defecated, the freshly defecated excretions were collected as did in captive monkeys. To avoid repeated specimen collections, monkeys were morphologically identified based on sex, body size, and pelage color. The baited-rope specimens and fecal swabs were kept in 1.5 mL of sterile lysis buffer at RT until DNA extraction. In total, 173 baited rope specimens and 204 fecal swab specimens were collected (see Table 2).

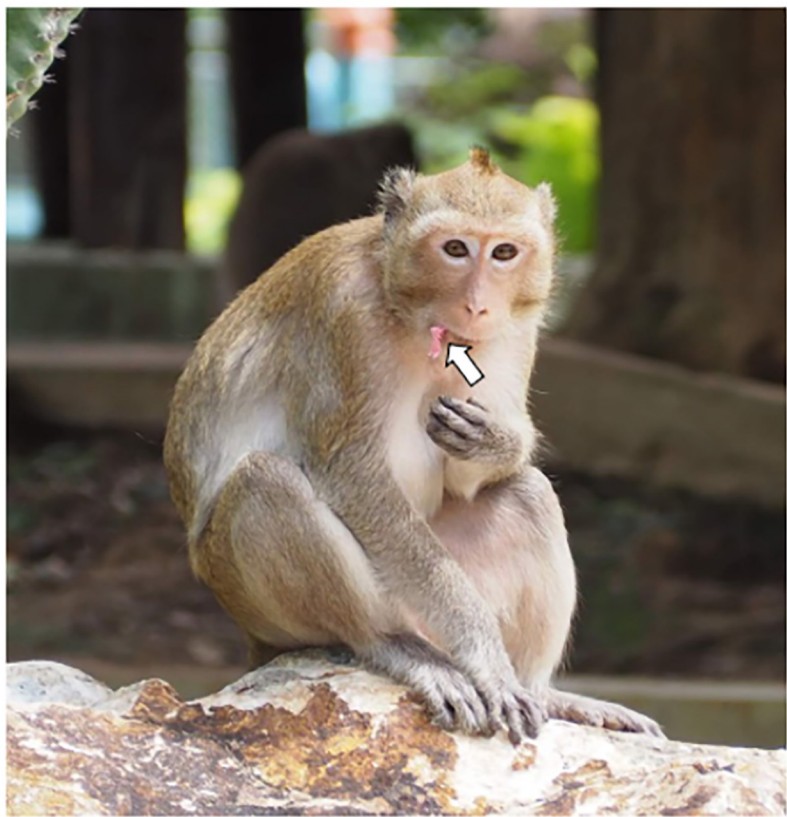

**Fig 1. Adult female common long-tailed macaque (*Macaca fascicularis fascicularis*) from the WKT population with a baited rope in her mouth (white arrow).**

The studies in captive long-tailed macaques at KBKWRC and in the four populations of free-ranging long-tailed macaques were approved by the Department of the National Parks, Wildlife and Plant Conservation of Thailand. The animal handling protocol was approved by the Institutional Animal Care and Use Committee of the National Primate Research Center of Thailand-Chulalongkorn University (NPRCT-CU; Protocol review number: 2075007). All methods were also performed in accordance with the relevant guidelines and regulations.

**Table 2. Detection of MTBCS in free-ranging common long-tailed macaques (*Macaca fascicularis fascicularis; Mff*) and Burmese long-tailed macaques (*M. f. aurea; Mfa*) from orally baited-rope and freshly dropped-fecal swab specimens.**

| Species | Location | Rope bait | | Fecal swab | |
|---|---|---|---|---|---|
| | | No. of specimens collected | No. of MTBCS-positive specimens (%; 95% CI) | No. of specimens collected | No. of MTBCS-positive specimens (%; 95% CI) |
| *Mff* | WKT | 104 | 5 (4.81; 0.7–8.9) | 112 | 8 (7.14; 2.4–11.9) |
| *Mfa* | TPK | 28 | 1 (3.57; 0.0–10.4) | 30 | 3 (10.00; 0.0–20.7) |
| | WWM | 24 | 2 (8.33; 0.0–19.4) | 40 | 5 (12.50; 2.3–22.7) |
| | MFRC | 17 | 1 (5.88; 0.0–17.1) | 22 | 2 (9.09; 0.0–21.1) |
| | Total | 173 | 9 (5.20; 1.9–8.5) | 204 | 18 (8.82; 4.9–12.7)* |

* indicates a significant difference (p < 0.05) from another specimen type.

### Statistical analysis

The standard prevalence rate and 95% confidence interval (CI) for differences in MTBCS prevalence in each macaque population were calculated using Chi-square statistical test. The correlation between MTBC prevalence and type of specimen collected (invasive and non-invasive) was analysed using Pearson correlation analysis. Data were analysed using IBM SPSS 28 for Mac (SPSS Inc., USA). Statistical significance was indicated at p-values of $\leq 0.05$.

## Results

### Specificity and sensitivity of MTBCS detection using IS6110 nested PCR

Comparison of the 181-bp nucleotide sequences of five MTB-positive monkeys (see Table 1), from invasive oral and fecal swab specimens, with the MTB H37Rv genome (NCBI reference sequence no. NC_000962.3) all showed a 100% sequence similarity (Fig 2). The BLASTn search with the NCBI-registered complete genomes indicated that the 181-bp nucleotide sequence obtained from the IS*6110* nested PCR could be specifically detected in the MTBC, including 574 strains of *M. tuberculosis*, eight strains of *M. bovis* BCG, three strains of *M. canettii*, two strains each of *M. bovis* and *M. africanum*, and one each of *M. caprae*, *M. microti*, and *M. orygis*. Thus, the IS*6110* nested PCR positive results in this study were interpreted as MTBCS-positive samples.

The optimized MTBCS nested PCR conditions of H37Rv DNA (ATCC27294) strain showed LOD values of 100 pg/µL in the first round PCR (580-bp PCR product) and 10 fg/µL in the second round (181-bp product) nested PCR in both the oral and fecal swab specimens (Fig 3A and 3B).

### Comparison of MTBCS frequencies between the invasive and non-invasive specimen collections in captive long-tailed macaques

For both the invasive and non-invasive oral and fecal specimen collections, the repetition of specimen collections (Day-1 and Day-5 for invasive specimen collections and Day-2 to Day-4 for non-invasive specimen collections) showed the same MTBCS results, and so only a single set of data for each type of specimens is reported in Table 1. For the invasive specimen collection, a positive PCR amplicon was detected in 4 out of 24 (or 16.67%) oral swabs and 1 out of 24 (or 4.17%) rectal swabs. Interestingly, the rope bait specimens and freshly defecated

```
Score:335 bits(181), Expect:4e-90,
Identities:181/181(100%),  Gaps:0/181(0%), Strand: Plus/Minus
```

**Fig 2. A 181-bp DNA sequence of *Mycobacterium tuberculosis* IS*6110* obtained from the nested PCR (Query) showing 100% sequence homology with a part of MTB H37Rv complete genome (subject; NCBI reference sequence no. NC_000962.3).**

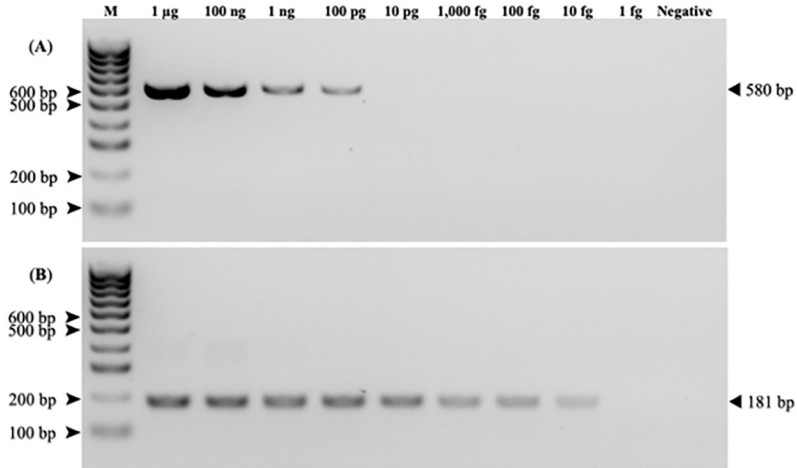

**Fig 3. The lower limit of detection of MTBCS in the first (580 bp, A) and the second (181 bp, B) round IS*6110*-nested PCR.** The PCR products were resolved and visualized in a 2% (w/v) agarose gel. M = Hyperladder 100 bp DNA ladder. Negative control is DEPC treated water.

excretion swabs showed 4 out of 24 (or 16.67%) and 3 out of 24 (or 12.50%) to be MTBCS positive, respectively. Thus, the results showed a significant correlation between the two types of oral specimens: oral swabs and baited ropes ($n = 24$, $r^2 = 1$, p-value $< 0.001$). However, there was no significant correlation between the two types of fecal specimens: rectal swabs and freshly defecated excretion swabs ($n = 24$, $r^2 = 0.30$, p-value $= 0.24$).

## Non-invasive specimen collections and MTBCS prevalence in free ranging long-tailed macaques

A significantly ($\chi^2 = 20.76$, $p = 0.01$) more frequent detection of MTBCS-positive monkeys was found using the fresh fecal swabs (8.82%; 95% CI; 4.9–12.7%) than the baited ropes (5.20%; 95% CI; 1.9–8.5%; Table 2). Positive PCR amplicons from the baited ropes were found in 9 out of 173 monkeys: five (4.81%) common long-tailed macaques and four (6%) Burmese long-tailed macaques, while that for the fresh fecal swabs was 18 out of 204 monkeys: eight (7.14%) common long-tailed macaques and 10 (11%) Burmese long-tailed macaques. There were no significant differences in the MTBCS-positive results between common and Burmese long-tailed macaques when using the baited ropes ($\chi^2 = 0.08$, p-value $= 0.77$) or the fresh fecal swabs ($\chi^2 = 0.87$, p-value $= 0.35$).

## Discussion

The transmission of severe infectious diseases from NHPs to humans has become a major scientific focus in recent years, especially in Africa and Asia–the habitat ranges of NHPs [26–28]. Thailand is located in southeast Asia where long-tailed macaques highly interface and interact with humans and infectious diseases, such as simian adenoviruses (SAdV-A, SAdV-B and SAdV-H) [29], SFV, hepatitis B virus, and *Plasmodium spp*. [7,30], have been reported in those animals. Moreover, in 2021 the WHO ranked Thailand as one of the countries with the highest human burdens of tuberculosis [31]. However, surveys of free-ranging long-tailed macaques living in Thailand [7] and the neighbouring Malaysia [32] did not detect MTBC with only the *Mycobacterium avium* complex being reported in Malaysian long-tailed macaques [32]. These

negative results might reflect that the diagnostic methods were not sensitive enough to detect a very low amount of the MTBC or the condition of the collected specimens was not suitable. Recently, surveillance for MTBCS in 26 populations of long-tailed macaques, totaling 1,647 animals, throughout Thailand detected MTBCS-positive monkeys in 13 populations (10 populations of common long-tailed macaques and three populations of Burmese long-tailed macaques) using IS6110 nested PCR [9]. However, invasive specimen collection methods (throat, buccal, and rectal swabs) are not practical for a large population survey and for a follow up of the MTBCS status in some monkey individuals.

The challenges of MTBC diagnosis in free-ranging NHPs are the sensitivity and specificity of the detection methods, the quality/quantity of collected specimens, the types of specimens, and the methods of specimen collections (invasive or non-invasive). To detect MTBC infection at the latent stage, interferon gamma release assay and antibody ELISA are mainly used. Thus, only invasively collected specimens from anesthetized animals, such as blood or plasma, can be used, which is impractical for wild/free-ranging animals. At the shedding active stage, direct smear microscopy (acid-fast bacilli), mycobacterial culture, and molecular testing (PCR, nested PCR, RT-PCR, GeneXpert, and Line probe assay) are available [33]. An advantage of detecting the active shedding stage is in predicting how the MTBC can be spread within or between populations and to other species in the vicinity. For a field study, rapid molecular testing is preferable because its sensitivity is comparable to the culture method, while the specificity can be improved based on the designed primers, and it is less-time consuming. The widely used methods are conventional PCR through the detection of the 16S rRNA gene [32] or multiplex PCR through detection of IS6110, 32-kD alpha protein (32-kDa), and MTP40 species-specific protein (mtp40) [7].

For specimen collection, non-invasive methods are preferable because it is difficult to catch free-ranging NHPs, especially non-habituated, wild animals. Previously, MTBC detection in non-invasively collected fecal samples was developed in chimpanzees and baboons [6]. Apart from the unpredictable defecation of animals, another disadvantage of MTBCS detection in fecal samples is that the pathogenic MTBCS DNA might be degraded when it passes through the gastrointestinal tract and becomes contaminated with intestinal microflora, and so conventional PCR might not be able to detect low levels of MTBCS in feces. Besides, if the animals do not have a wet cough and swallow their sputum back into the gastrointestinal tract, the determination of MTBCS in feces is impractical.

In this study, the previously developed rope bait method [17] was modified for oral specimen collections from four previously reported MTBC-infected populations of free-ranging long-tailed macaques [9]. The specimens (oral rope bait and fecal swab) were preserved in sterile lysis buffer and can be stable at RT for a year [20,21]. Compared to conventional PCR, the use of nested PCR with two primer sets in this study improved the sensitivity, with a LOD as low as 10 fg/ml in the second round of PCR in both types of specimens. As seen in Fig 3, the LOD was 100 pg in the first round of PCR, some 10,000 times higher than in the second round. This might explain why the previous studies using only conventional PCR [7,32] did not detect MTBC. In our study, a series of specimen collections in a single animal during a short period of time (Day-1 and 5 for invasive specimen collection and Day-2 to Day-4 for non-invasive specimen collection) were done because we wanted to confirm that the MTBCS negative results were not by the technical errors of specimen collection. Since all repetitive collected specimens showed the same results throughout the 2 and 3-day periods, this confirmed that the negative results were not the specimen collection error, but the animals were no MTBCS.

Although we proposed the rope bait method for obtaining oral specimens for MTBCS detection, this method has some disadvantages to bear in mind. The proportion of positive

MTBCS samples in the non-invasively collected oral specimens (rope bait) was lower than that in the non-invasively collected fecal specimens (fecal swab), which may reflect that the animal did not chew the baited rope for long enough before discarding it onto the ground [17,18]. The discarded baited ropes after chewing may be contaminated with organic as well as inorganic substances or other pathogens on the ground that could interfere with the PCR [34,35]. Additionally, the rope-bait method cannot be used in non-habituated free-ranging or wild macaques that are unwilling to allow humans to approach. In addition, in some habitat types, such as mangrove forests, islands, or high cliffs, it is difficult to retrieve the discarded baited rope.

Another problem of the rope bait method for socially organized macaques is that if many animals stay in the same place, the higher rank animals always possess the sugar-baited ropes. In this case, fecal sample collection is suggested via aggressive food intake competition between macaques [36]. As seen in our results, the non-invasively collected freshly defecated excretion swabs showed a higher frequency of MTBCS-positive cases than the invasively collected rectal swabs, which might be because a direct fecal swab could acquire a higher amount of the shed microbe than a rectal swab. Since there were no significant differences in MBTCS-positive detection levels between the oral rope bait and fecal swab in each macaque population, we would suggest using both methods for specimen collection, depending on the circumstance of the field study.

In conclusion we present the feasibility of large-scale, non-invasive specimen collection from free-ranging NHPs for MTBCS detection using IS*6110* nested PCR. This technique also provides an opportunity for the monitoring of other endemic pathogen diseases that infect the respiratory tract to avoid the requirement for a high financial, manpower, and time investment.

## Supporting information

**S1 Raw images.**
(TIF)

## Acknowledgments

We are grateful to Kritthapron Khumnonchai and Chakkaphong Kongchanda for supporting and participating in KBKWRC, Chachoengsao province, eastern Thailand. We thank all the officers and staff of the Department of the National Parks, Wildlife and Plant Conservation, Thailand for permission to conduct this research.

## Author Contributions

**Conceptualization:** Suchinda Malaivijitnond.

**Formal analysis:** Suthirote Meesawat.

**Funding acquisition:** Suchinda Malaivijitnond.

**Investigation:** Suthirote Meesawat, Nalina Aiempichitkijkarn, Mutchamon Kaewparuehaschai.

**Project administration:** Suthirote Meesawat.

**Resources:** Mutchamon Kaewparuehaschai, Suchinda Malaivijitnond.

**Supervision:** Saradee Warit, Suchinda Malaivijitnond.

**Visualization:** Suthirote Meesawat, Suchinda Malaivijitnond.

**Writing – original draft:** Suthirote Meesawat, Suchinda Malaivijitnond.

**Writing – review & editing:** Suthirote Meesawat, Saradee Warit, Suchinda Malaivijitnond.

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
