## [Decision Letter · Decision Letter 0]

5 Jul 2023

PONE-D-22-34908Non-invasive specimen collections for Mycobacterium tuberculosis detection in free-ranging long-tailed macaques (Macaca fascicularis)PLOS ONE

Dear Dr. Malaivijitnond,

Thank you for submitting your manuscript to PLOS ONE. After careful consideration, we feel that it has merit but does not fully meet PLOS ONE’s publication criteria as it currently stands. Therefore, we invite you to submit a revised version of the manuscript that addresses the points raised during the review process.

Please see below the comments and suggested MINOR revisions made by the individual(s) who reviewed your manuscript.  If provided, the referee's report(s) indicate the revisions that need to be made before it can be accepted for publication.

We look forward to receiving your revised manuscript.

Kind regards,

Ricardo Santos

Academic Editor

PLOS ONE

Journal Requirements:

Reviewers' comments:

Reviewer's Responses to Questions

**Comments to the Author**

1. Is the manuscript technically sound, and do the data support the conclusions?

Reviewer #1: Yes

2. Has the statistical analysis been performed appropriately and rigorously? 

Reviewer #1: I Don't Know

3. Have the authors made all data underlying the findings in their manuscript fully available?

Reviewer #1: Yes

4. Is the manuscript presented in an intelligible fashion and written in standard English?

Reviewer #1: Yes

5. Review Comments to the Author

Reviewer #1: The authors presented quite a significant problem related to the collection of diagnostic material from free-living animals, especially those belonging to the group of primates. Due to the numerous restrictions related to the violation of the welfare of free-living animals and the reduction of their stress, and, on the other hand, the need to quickly identify zoonotic pathogens, the non-invasive method of sampling and its use in MTBC diagnostics seems very applicable and I certainly recommend the article for publication. However, a few clarifications still need to be made, even though the manuscript has already undergone initial revisions:

It is not clear why the material was taken again after 5 days using the invasive method. Isn't this too short a time for animals to fully recover after pharmacological treatment? Please justify such a short time as well as repeating the collection with a non-invasive method on the 2nd, 3rd and 4th day.

I also believe that the method of collecting material from free-living animals populations is described too generally. Animal populations were studied, but in what space? It is hard to imagine that, as in the case of animals kept in cages, faeces were collected immediately after defecation or a rope was found with the certainty that it had been chewed by the animal - please specify this part of the experiment.

Were the results of the research supported by microbiological culture or were they based only on the results of molecular tests? If not, please explain in the text why.

6. PLOS authors have the option to publish the peer review history of their article (what does this mean?). If published, this will include your full peer review and any attached files.

Reviewer #1: **Yes: **ANETA NOWAKIEWICZ

---

## [Author Response · Author response to Decision Letter 0]

7 Jul 2023

PLOS ONE

Manuscript No. # PONE-D-22-34908

Manuscript title: Non-invasive specimen collections for Mycobacterium tuberculosis detection in free-ranging long-tailed macaques (Macaca fascicularis)

Submission as Research article

Authors: Suthirote Meesawat, Nalina Aiempichitkijkarn, Saradee Warit, Mutchamon Kaewparuehaschai, Suchinda Malaivijitnond 

Response to the Editor in Chief

Thank you very much for all of your comments to make this manuscript better. The manuscript has been thoroughly revised as indicated with yellow-highlighted.

Reviewer: 1

The authors presented quite a significant problem related to the collection of diagnostic material from free-living animals, especially those belonging to the group of primates. Due to the numerous restrictions related to the violation of the welfare of free-living animals and the reduction of their stress, and, on the other hand, the need to quickly identify zoonotic pathogens, the non-invasive method of sampling and its use in MTBC diagnostics seems very applicable and I certainly recommend the article for publication. However, a few clarifications still need to be made, even though the manuscript has already undergone initial revisions:

It is not clear why the material was taken again after 5 days using the invasive method. Isn't this too short a time for animals to fully recover after pharmacological treatment? Please justify such a short time as well as repeating the collection with a non-invasive method on the 2nd, 3rd and 4th day.

Ans: For clarification, these animals did not receive any pharmacological treatment, they were only subjected to biological specimen collection for MTBCS detection. The invasive specimen collection was done twice on Day-1 and Day-2, and non-invasive specimen collection was done thrice on Day-2, 3 and 4. These multiple collections in a single animal during a short period of time were done because we wanted to confirm that there were no technical errors of specimen collection on MTBCS negative results. Since all repetitive collected specimens (both invasive and non-invasive methods) showed the same results throughout the 2 and 3-day periods, this indicates that the negative results were not the specimen collection error, but the animals were no MTBCS. 

Note, this explanation was also added in the Discussion (Page 16 and 17) of the revised manuscript.

I also believe that the method of collecting material from free-living animals populations is described too generally. Animal populations were studied, but in what space? It is hard to imagine that, as in the case of animals kept in cages, faeces were collected immediately after defecation, or a rope was found with the certainty that it had been chewed by the animal - please specify this part of the experiment.

Ans: The habitat type of each population and the GPS of each location were added (Page 10). The procedures of baiting the 20% sucrose ropes to animals and collecting of the discarded chewed rope were added into the manuscript. The procedure of following monkey before collection of the freshly defecated excretions was also added (Page 10 and 11).

Were the results of the research supported by microbiological culture or were they based only on the results of molecular tests? If not, please explain in the text why. 

Ans: The results were based only on the molecular test, and we did not perform the culture for the following reasons.

 i)For fecal specimen: Although the fecal swab was done in the inner part of the feces and had low possibility to be contaminated with ground pathogens, it can not be used for culture because it was highly contaminated with gut microflora. 

ii)For oral specimen: The amount of MBTCS is too low for culture and it was contaminated with other pathogens on the ground that could interfere with the growth of the MTBC.

We did not add this explanation in the manuscript because we thought that our writing in the introduction and methods was clear enough to convey our intention that we aimed to use only IS6110-nested PCR method for MTBCS detection. We are also afraid that adding the explanation why we did not confirm the nested-PCR results by microbiological culture will cause confusion to the readers.

---

## [Decision Letter · Decision Letter 1]

31 Jul 2023

Non-invasive specimen collections for Mycobacterium tuberculosis detection in free-ranging long-tailed macaques (Macaca fascicularis)

PONE-D-22-34908R1

Dear Dr. Malaivijitnond,

We’re pleased to inform you that your manuscript has been judged scientifically suitable for publication and will be formally accepted for publication once it meets all outstanding technical requirements.

Kind regards,

Ricardo Santos

Academic Editor

PLOS ONE

Additional Editor Comments (optional):

Reviewers' comments:

Reviewer's Responses to Questions

**Comments to the Author**

1. If the authors have adequately addressed your comments raised in a previous round of review and you feel that this manuscript is now acceptable for publication, you may indicate that here to bypass the “Comments to the Author” section, enter your conflict of interest statement in the “Confidential to Editor” section, and submit your "Accept" recommendation.

Reviewer #1: All comments have been addressed

2. Is the manuscript technically sound, and do the data support the conclusions?

Reviewer #1: Yes

3. Has the statistical analysis been performed appropriately and rigorously? 

Reviewer #1: Yes

4. Have the authors made all data underlying the findings in their manuscript fully available?

Reviewer #1: Yes

5. Is the manuscript presented in an intelligible fashion and written in standard English?

Reviewer #1: Yes

6. Review Comments to the Author

Reviewer #1: I have no any other comments. I recommend the manuscript for publication in its current form because the authors have taken into account all my comments. In its current form, the manuscript sounds good and clearly describes a non-invasive methodology for collecting material from free-living animals, which may have an application character. This enables potential monitoring of the occurrence of zoonotic diseases, which is extremely important in terms of assessing the risk to public health

7. PLOS authors have the option to publish the peer review history of their article (what does this mean?). If published, this will include your full peer review and any attached files.

Reviewer #1: **Yes: **Aneta Nowakiewicz

---

## [Editor Report · Acceptance letter]

15 Aug 2023

PONE-D-22-34908R1 

Non-invasive specimen collections for *Mycobacterium tuberculosis* detection in free-ranging long-tailed macaques (*Macaca fascicularis*) 

Dear Dr. Malaivijitnond:

I'm pleased to inform you that your manuscript has been deemed suitable for publication in PLOS ONE. Congratulations! Your manuscript is now with our production department. 

Kind regards, 

on behalf of

Dr. Ricardo Santos 

Academic Editor

PLOS ONE